# Prolonged Gel Delivery to Oral Cavity from a Silicone Tube: In Vivo Assessment

**DOI:** 10.3390/pharmaceutics17091095

**Published:** 2025-08-22

**Authors:** Suhail Alghanem, Ewelina Dziurkowska, Mateusz Lampkowski, Iwona Ordyniec-Kwaśnica, Małgorzata Sznitowska

**Affiliations:** 1Department of Pharmaceutical Technology, Faculty of Pharmacy, Medical University of Gdansk, 80-210 Gdansk, Poland; suhail.alghanem@gumed.edu.pl; 2Department of Analytical Chemistry, Faculty of Pharmacy, Medical University of Gdansk, 80-210 Gdansk, Poland; ewelina.dziurkowska@gumed.edu.pl; 3Department of Dental Prosthetics, Faculty of Medicine, Medical University of Gdansk, 80-210 Gdansk, Poland; mateusz.lampkowski@gumed.edu.pl (M.L.); iwona.ordyniec-kwasnica@gumed.edu.pl (I.O.-K.)

**Keywords:** silicone tubes, hydrogel, erosion, prolonged release, medical device, comfort

## Abstract

**Objectives**: This study evaluated the comfort of using silicone tubes installed in the oral cavity as a reservoir for a hydrogel that allows for a slow delivery of the active substance acting locally or systemically. **Methods**: Perforated silicone tubes 8 cm long with two internal diameters were used: T1 (1.5 mm) and T2 (2.4 mm). The reservoirs were filled with hydrogel placebo formulations: carbomer 1.5% (C), hydroxyethylcellulose 4% (HEC), or hydroxypropylmethylcellulose (hypromellose) 3% (HPMC). Physical parameters of the gel were determined with a viscometer and a texture analyzer. During 4 h of application, the volunteers reported sensory perceptions, and the rate of gel erosion was evaluated. The results were correlated with the viscosity, rheology, and dissolution rate of the gels measured in vitro. **Results**: Volunteers reported only mild discomfort wearing the device, preferring smaller-sized tubes. The tubes were easy to apply and generally comfortable, with no reports of significant discomfort. Despite similar viscosity and rheology, the polymer type had a significant impact on erosion rate, both in vitro and in vivo. After 4 h of application in vivo, more than 90% of the carbomer gel remained in the tube, while in the case of less cohesive HPMC or HEC gels, this was about 50%. A statistically significant correlation was observed between the in vitro and in vivo mean erosion percentages for the HEC and HPMC gels. **Conclusions**: This study supports the use of silicone tubes as effective reservoir devices for prolonging the residence time of drug formulations in the oral cavity.

## 1. Introduction

Oral administration is the most convenient method of treatment for achieving both local and systemic effects, offering high patient compliance. Administering a drug into the mouth may be done with the aim of swallowing it immediately or keeping it in the mouth for as long as possible, which applies primarily to drugs acting locally, e.g., in sore throats or dental diseases. The formulations for local drug delivery are available in the form of solids (lozenges, patches), semisolids (gels), or liquids (sprays, solutions, mouthwashes). However, these formulations face several limitations, including poor retention at the application site due to salivation along with swallowing, chewing, and phonation, which quickly washes away most of the drug from the site of application, leading to short residence time in the oral cavity, even if some mucoadhesion is acquired. This results in reducing the therapeutic efficacy, inconsistent drug dosing, and delivery. Issues such as the need for taste masking or involuntary swallowing further impact the effectiveness and patient compliance. To maintain a therapeutic level of the active substance in the oral cavity, the patients should administer the preparation several times throughout the day, but usually with gaps in treatment during sleep. To overcome these challenges, different medical devices were developed [1,2,3,4,5,6], but their use in clinical practice is unnoticeable, mostly because of the patients’ discomfort, which is caused by the size of the devices and the hard materials used in their construction. 

Mouthguards have long been used in dentistry for protection during sports or orthodontic realignment. Advancements in technology have opened up new possibilities for utilizing these devices in therapeutic applications, particularly for drug delivery within the oral cavity. Biocompatible materials are essential for ensuring the safe administration of medications into the oral cavity [7,8,9,10,11]. Silicone is a polymer widely used in the biomedical field due to its excellent mechanical properties and biocompatibility. It has been used in applications such as medical inserts, contact lenses, urinary catheters, and prosthetics. For example, the silicone-based device, Bocaliner™, prolongs the retention of oral topical therapies such as benzydamine mouthwash [6]. Advances in 3D printing technology have opened new possibilities for silicone in medical devices, offering customizable solutions for medical applications [12].

Despite these innovations, current drug-delivery mouthguards or other reservoir-type medical devices face limitations in terms of long-term use, reusability, cost, and drug release consistency. For example, Bocaliner™ is a rigid silicone-based insert designed to enhance topical therapy in conditions such as oral mucositis by reducing saliva dilution and passively keeping medications in contact with the oral mucosa after administration. However, it does not serve as a reservoir for controlled drug release [6]. Eluting mouthguards, on the other hand, typically involve either an internal coating of sustained-release active pharmaceutical ingredient (API) varnish or the incorporation of APIs into polymeric filaments, often produced using 3D printing [8,11]. 

In contrast, our proposed system (silicone tube) uses a soft, flexible, refillable, and replaceable internal reservoir filled with a formulation that elutes through orifices over time, serving as both a barrier and a release platform. The design allows for adjusting residence time, refilling options, and potential integration with a custom-fitted mouthguard, which improves comfort, safety, and sustained delivery. This reservoir concept, unlike existing rigid solutions, introduces a platform for prolonged local or systemic intraoral drug administration. Moreover, our simple system can be integrated into a custom-designed mouthguard. The reservoir, consisting of a silicone tube with orifices, acts as a physical barrier that shields the formulation from being prematurely washed away by saliva, allowing for prolonged drug release. In this study, we evaluate the patient comfort and in vivo erosion time of the hydrogel formulations within this reservoir, aiming to improve the efficacy and usability of oral drug-delivery devices. 

## 2. Materials and Methods

### 2.1. Reservoir Design

Figure 1 presents the tubes used in this study. The reservoir consists of a silicone tube (8 cm in length) with 4 orifices (1.5 mm in diameter). Two sizes of tubes were tested: with an internal diameter of 1.5 mm (T1) and 2.4 mm (T2), and an external diameter of 3 mm and 4 mm, respectively. Both ends of the tubes were closed.

The tubes were filled with sterile gel. The T1 capacity was 140 mg, and the T2 capacity was 350 mg of the gel.

### 2.2. Hydrogel Preparation and Determination of Rheology and Texture

The gels were prepared by dissolving the following polymers in water: hypromellose 3% *w*/*w* (HPMC, Metolose 60 SH-10000, ShinEtsu, Tokyo, Japan), hydroxyethylcellulose 4% *w*/*w* (HEC, Natrosol 250 M, Ashland, Wilmington, DE, USA), or carbomer 1.5% *w*/*w* (C, Carbopol 974, Lubrizol, Rotterdam, The Netherlands). Propylene glycol 20% *w*/*w* (POCh, Gliwice, Poland), mannitol 10% *w*/*w* (POCh, Gliwice, Poland), menthol 0.05% *w*/*w* (Farm-Impex, Gliwice, Poland), and pigment (0.1% blue brilliant) were the other excipients used. The gels were sterilized in an autoclave.

Rheology and viscosity of gels were determined using a Julabo F12 thermostat (Seelbach, Germany) and a Viscometer VT 550 with a cone-plate geometry (Thermo Fisher Scientific, Schwerte, Germany), and adhesiveness was analyzed by a texture analyzer (TA.XT-Plus Stable Micro System, Godalming, UK). The measurements were performed at room temperature, and the results were presented as the mean ± standard deviation (SD) based on 3–5 different measurements. The rheograms were obtained by plotting shear stress versus shear rate (0–100 s−^1^), and the dynamic viscosity at shear rate 20 s^−1^ was determined. Hardness, cohesion, and adhesion measurements were performed by recording the forces appearing when immersing in the gel a disc probe (40 mm in diameter) at a depth of 5 mm and withdrawing it at a speed of 2 mm/s. This method was adapted from previously published protocols by Wroblewska et al. [13] and Hurler et al. [14].

### 2.3. In Vitro Hydrogel Erosion Study

The silicone tube filled with a gel was placed in 100 mL of artificial saliva, stirred (500 rpm) on a magnetic stirrer (Heidolph, Schwabach, Germany) at 37°C. Every 30 min, the tubes were removed from the medium, and photos were taken to evaluate the gel elution. This was repeated until the hydrogel was fully eluted. Each formulation was tested in triplicate.

Artificial saliva was prepared following the guidelines of the German Institute for Standardization (DIN) [15,16]. The composition of 1000 mL was as follows: 0.68 g of potassium hydrogen phosphate, 0.33 g of sodium chloride, 0.17 g of magnesium chloride hexahydrate (POCh, Gliwice, Poland), 0.15 g of calcium chloride dihydrate (Chempur, Piekary, Poland), 0.75 g of potassium chloride (Bioskamed, Warsaw, Poland), and 0.53 g of potassium carbonate (Fluka Chemicals, Buchs, Switzerland). The final pH of the solution was adjusted to 6.8 ± 0.1.

### 2.4. In Vivo Study

This study was conducted based on the approval of the Bioethics Committee at the Medical University of Gdansk. Six volunteers (two men and four women) were recruited (ages 24–47 years). Every volunteer examined three formulations (different tubes and types of polymer) on three different days. Each formulation in one of the tubes was examined by 3 volunteers. The site of application was the gum upper and the teeth, and the volunteers were asked to keep the tube in this site for 4 h. Every 30 min, the tube was removed for a while to take a photo of the remaining gel. The elution of the gel was evaluated as the visually estimated percentage of the tube length still filled with the colored gel. 

The volunteers were asked to complete a questionnaire. They rated the acceptability of the tube’s length and size on a scale from 1 “difficult” up to 4 “easy”, and comfort at various time points using a scale from 0 (“feel nothing”) to 3 (“uncomfortable”). They also rated whether the tube applied interfered with their ability to speak (1—“not at all” and 5—“I could not speak”). 

Taste perception and cold sensation were evaluated using a structured questionnaire. Participants were asked to report whether they experienced any taste (and to describe it if possible) and whether they perceived any cold sensation at the application site of the tube. Assessments were conducted at multiple time points throughout the 4-h application period. This approach allowed for dynamic tracking of sensory experiences over time and provided qualitative data on the intensity of taste and cold sensations associated with each gel formulation.

Three volunteers were asked to test the three different hydrogels (350 mg) at the same application site without using tubes. They were asked to evaluate the residence time, taste, and sensation experienced during the application period. 

### 2.5. Statistical Analysis 

Statistical analysis of the results was performed using descriptive statistics, a two-way ANOVA test, a post hoc Tukey test, and Pearson’s correlation coefficient. All statistical computations were carried out in Excel (version 2013 (15.0.5589.1000) and Python (version 3.12.11) using Google Colab. 

Both in vitro and in vivo erosion percentages of HEC and HPMC hydrogels were plotted to visualize their correlation. Pearson’s correlation coefficient was calculated.

## 3. Results

### 3.1. Characteristics of Hydrogels

Table 1 shows the physical characteristics and pH of the placebo hydrogels. Figure 2 presents the results of the rheological test. No tixothropy of the gels was noted. Since different types of polymers resulted in various viscosities of the solutions, based on the preliminary work for this project, the appropriate concentrations were optimized to maintain a standardized viscosity (~20 Pa·s) suitable for practical handling (filling the tube using a syringe). Moreover, the elimination of differences in viscosity allowed us to explore other structure–function relationships, although the viscosity was similar, but the parameters measured with a texture analyzer were different. The carbomer gel (C) showed similar adhesiveness but larger hardness and cohesiveness in comparison to hydrogel formulations prepared with cellulose derivatives HEC or HPMC (Table 1).

### 3.2. In Vitro Hydrogel Erosion Study

As presented in Figure 3, the in vitro erosion time lasted around 270 min for the HEC gel in both tubes T1 and T2. This time was 360 min for the HPMC gel in tube T1 and 330 min for the HPMC gel in tube T2. The carbomer gel exhibited an erosion time of 480 min in both tubes, with not more than 15% erosion after 4 h and less than 35% at the 450 min time point. It was challenging to determine the exact erosion percentage of the carbomer gel every 30 min, as changes during 450 min were minimal and difficult to quantify. Surprisingly, at the time point of 480 min, all the carbomer gel disappeared, indicating that during the last 30 min, a fast degradation of the carbomer viscous structure took place. The observation was reproducible for three samples and both tube sizes. One of the possible explanations for such a phenomenon may be the diffusion of calcium and magnesium ions from the saliva-mimicking solution, so that when a certain concentration of calcium and magnesium is reached in the gel filling the tube, the gel loses viscosity due to interactions with divalent ions and is easily washed out. Among the polymers tested, only carbomer is ionic in character and can readily react with cations [17].

The results of the two-way ANOVA test showed a highly significant effect of polymer type on in vitro erosion time (*p*-value = 4.38 × 10^−11^, less than 0.001), whereas the tube diameter had no significant effect (*p*-value = 0.175 > 0.05). Furthermore, no significant interaction between polymer type and tube size was observed (*p* = 0.168, *p* > 0.05). The longest time of the gel residence in the tube, despite an intensive mechanical stress (stirring), was observed for the carbomer gel, and no difference was observed depending on the amount of gel, related to the tube size. In contrast, the HEC and HPMC gels eroded significantly faster, with only minor variations attributable to tube diameter. This effect indicates that the gel rheology is not so important in predicting the rate of its elution from the tube, but this should be correlated with texture parameters such as hardness and cohesion (Table 1).

### 3.3. In Vivo Comfort

The participants reported that a single-time explanation from the researchers was enough to correctly apply the tube; they also found it easy to apply the tube in the oral cavity. Most participants were able to keep the tubes in place at the current application site, although one volunteer noted some movement during the experiment. Three out of six volunteers would prefer the lower gum as an alternative application site. All participants preferred smaller-sized tubes. However, four participants rated the comfort level as “Mild” (they felt it, but it was not disturbing), while the other two participants described it as “Tolerable” (they felt it as disturbing). None of the volunteers rated either tube as "Uncomfortable”.

All the participants confirmed that both tubes, T1 and T2, did not interfere with their daily activities, such as walking, working on the computer, using a mobile phone, or performing tasks. When volunteers were asked to evaluate how the tubes affected their ability to speak, four of the participants reported a slight or moderate hindrance, while the other two participants indicated a stronger impact. No participants reported being completely unable to speak with either tube.

### 3.4. Taste and Sensation Evaluation

Since the gels were flavored with menthol and mannitol, the taste sensations could indicate the time of gel elution from the tube. For formulation with carbomer, no taste or cold sensation was reported with tube T1, while in the case of the larger tube T2, a sweet taste was reported by two out of three participants; however, with no cold sensations reported. For the HEC and HPMC gels, a sweet taste was reported by most participants, and two of them reported a mint taste. A cold sensation was experienced only by two of the participants who applied the HPMC gel, with no relationship to the tube size.

When asked if they would be willing to replace more frequent conventional treatments (such as gels, mouthwashes, and lozenges) with the long-lasting tube for local treatment in the oral cavity, three of the volunteers expressed a preference for the tube device, while the other three were uncertain. Additionally, four of the participants reiterated their preference for smaller or flatter tubes when they were asked if they had any further comments about the use of the tube, its application, or the device overall.

### 3.5. In Vivo Hydrogel Erosion Study

When the gels were applied in a typical manner, without tubes, all the participants experienced a cold sensation after application. The taste was described as mint by two of the participants, while the third one indicated it was hard to specify the taste. The total elution times ranged from 5–10 mins for the HPMC and HEC gels to 10–15 mins for the carbomer formulation. 

The in vivo study with the gels applied in tubes lasted for 4 h. During this time, the percentage of hydrogel remaining in the tube could be monitored visually as the gels were colored. Figure 4 illustrates the HPMC hydrogel remaining in the tube (T2) after 4 h of application. In this example, the in vivo gel elution was estimated at approximately 30%, based on the observed empty spaces within the tube. 

Quantitative analysis of erosion occurring both in vivo and in vitro was assessed visually by measuring the length of tube segments not stained blue (Figure 4). A digital evaluation of the photos using an image analysis program (open source ImageJ program, version 1.54g) was also applied, yielding extremely similar results (see Appendix A).

Figure 5 demonstrates the portions of the gels remaining in the tubes after 4 h of application, expressed as a percentage of the total dose. Formulation with carbomer showed the lowest erosion rates, and the size of the tubes was not relevant: after 4 h, over 90% of the gel was still present in the device. Approximately 50% of the HPMC gel was eluted in the same time, with only a small difference regarding the tube size. In the case of these two gels, the differences were not large when the results from three volunteers receiving the same formulations were compared. The results were not so clear in the case of the HEC gel because individual differences were significant—depending on the person receiving the gel, the amount of the gel eluted after 4 h was in the range 20–100%. More statistical analysis of in vivo erosion (%) results - confidence intervals available in Appendix A.

The data were statistically analyzed using a two-way ANOVA test. The type of formulation significantly influenced the erosion behavior of the hydrogels (*p* = 0.0095). In contrast, the subject factor, representing individual variability among the volunteers, showed *p* > 0.05, indicating no statistically significant effect on the results.

Post hoc analysis using Tukey’s Honestly Significant Difference (HSD) test revealed that the T1-HEC formulation had significantly higher erosion compared with both T1-C (*p* = 0.0339) and T2-C (*p* = 0.0313). No other pairwise differences were statistically significant. These findings highlight the primary role of formulation type in determining erosion characteristics.

### 3.6. In Vivo–In Vitro Correlation

The in vivo elution percentage for HEC and HPMC gels in both tubes T1 and T2, measured every 30 min, was compared to the in vitro elution percentage obtained at the same time points in artificial saliva (pH 6.8). These data were plotted and are shown in Figure 6. A strong correlation between the in vivo and in vitro results was demonstrated for HPMC: the Pearson’s correlation coefficients calculated for T1_HPMC and T2_HPMC were 0.9782 (*p*-value = 1.33 × 10^−6^) and 0.970 (*p*-value = 4.78 × 10^−6^), respectively. However, as previously mentioned, in the case of the HEC gel, larger interindividual in vivo differences occurred and the in vivo–in vitro correlation was not so clear, especially in the T1 tubes: the Pearson’s correlation coefficients calculated for T1_HEC, T2_HEC were 0.9037 (*p*-value = 0.000434) and 0.985 (*p*-value = 3.35 × 10^−7^), respectively. This indicates that the observed correlations are statistically significant and unlikely to be due to random chance. 

## 4. Discussion

This study demonstrates that silicone tubes effectively extend the residence time of hydrogel formulations at application sites within the oral cavity, while also offering acceptable user comfort. These findings validate the potential of silicone tubes as reservoirs for personalized medical devices, such as mouthguards, to enhance treatment safety and effectiveness, particularly for therapies administered during sleep. In addition, the concept of this simple replaceable/refilling reservoir will open the door to using the device for many medical situations, locally or systemically. Among the potential applications of the device is the treatment of dry mouth, which needs prolonged release of the medicine inside the oral cavity. By integrating the proposed silicone reservoir into a customized mouthguard, the treatment can become more comfortable and safer by preventing accidental swallowing of the reservoir. This is especially beneficial for nighttime therapies. 

The hydrogel formulations were developed with polymer concentrations optimized to achieve a dynamic viscosity of approximately 20 Pa·s, which is suitable for non-problematic introduction of the gel into the tube using a syringe with an 18 G (1.2 mm) needle. Texture profile analysis provides critical insights into hydrogel’s response to external forces. This analysis is particularly useful for predicting how samples behave under physiological conditions, such as the stress experienced during administration, and for evaluating the ease of removing hydrogel formulations from the reservoir during in vivo application. Hydrogels must possess suitable mechanical properties, including hardness, adhesiveness, and cohesiveness [13,14,18].

The formulations demonstrated similar viscosity, allowing a direct comparison of their mechanical properties. Textural analysis showed that the carbomer hydrogel exhibited superior hardness, adhesiveness, and cohesiveness compared to HEC and HPMC. Consistent with these textural properties, the carbomer formulation displayed lower erosion rates in both in vitro and in vivo environments. These findings highlight the role of the type of polymer on hydrogel erosion behavior. Therefore, achieving the optimal textural and rheological properties of hydrogels is essential to meet specific therapeutic requirements.

For the in vitro erosion study, the results agree with the textural properties of the hydrogels, where the carbomer formulation required more time to erode under the same conditions. Under the investigated conditions, no differences in erosion time regarding the internal diameter of the tube (T1 or T2) for the same formulations were observed, which suggests that the type of polymer played the most important role in the erosion time, and this is also true under in vivo conditions. It is clear that the residence time of the gel in the device and the drug release time may be controlled by the choice of gel-forming polymer. In addition to the choice of polymer, the design of the reservoir plays a critical role in controlling drug release. In this study, tubes were designed with four orifices, each with a diameter of 1.5 mm, but changing the number, shape, or size of the orifices could further refine the drug release profile, enabling more precise control over the release rate. 

In our current design, the orifices were initially oriented in a position to promote directional flow toward the mucosal surface. However, due to the flexibility of the silicone tube and natural oral movements during application, the device did not consistently maintain the same position. Since our aim is not to propose a device for close and fixed contact with the mucosa in order to obtain direct drug absorption, the orientation of the orifices is irrelevant: the gel and API are supposed to be delivered to saliva. However, incorporating the reservoir into a custom-fitted mouthguard could help better control the orifice orientation and directional flow to the mucosa. 

The erosion rates of hydrogels in vitro and in vivo exhibit significant differences due to environmental and operational factors. In vitro, the erosion was measured under controlled conditions, leading to reproducible results across the formulations. In vivo, however, the erosion rates were influenced by additional factors, including physiological variability among participants, the dynamic nature of oral movements (mechanical stress)—which subjected the tube to pressure, causing hydrogel release through this mechanism—and variations in saliva flow and composition [19,20]. 

Despite these differences, this study demonstrated that the proposed simple in vitro model could reliably predict the in vivo elution behavior of the gel formulations in the tested tubes. The predictive capability of the in vitro model supports its potential utility as a valuable tool for optimizing gel formulations and reservoir designs before in vivo testing. From a pharmaceutical development perspective, such predictive models are valuable for both formulation optimization and regulatory justification, especially in the case of drug–device combination products, where demonstrating a reliable in vitro–in vivo relationship is critical for approval. This correlation can be beneficial to reduce the burden of in vivo studies during early development of the formulations and the design of the device.

We acknowledge the limitations associated with the small number of samples tested in vivo, but this study was intended to establish proof-of-concept, with the aim of focusing on the evaluation of the device’s usability and the in vivo behavior of the gel carrier. While the number of human participants and trials was small, it was sufficient to meet the preliminary aims of assessing device comfort, functionality, and formulation erosion time. These initial results provide a strong basis for designing more comprehensive future studies involving active pharmaceutical ingredients (APIs), with broader participant groups. 

The evaluation of taste and sensation revealed that taste recognition during tube application was limited, with gels formulated with cellulose derivatives (HEC and HPMC) exhibiting stronger taste recognition compared to carbomer. This weak sensation of menthol taste is the effect of its slow release during tube application. This finding is particularly significant for the potential use of bitter-tasting drugs in the device, as it may reduce the need for taste masking. 

Comfort is a key consideration in the design of devices proposed for patients. Feedback from volunteers in this study highlighted the importance of designing the reservoir to fit the gum anatomy above the teeth, suggesting that a flat design rather than a round one may enhance comfort. Customization, guided by dental professionals, could enhance fit and reduce irritation, ensuring the device suits the patient’s oral anatomy. Dentists, therefore, would play a crucial role in tailoring the design, fitting, selecting proper materials for fabrication, and patient education about the device. Two participants reported a strong impact on their ability to speak. We believe that designing the tube based on the volunteers’ feedback to better fit the anatomy of the upper or lower gums will increase patient comfort and improve their ability to speak, particularly for long-term use and overnight use to avoid sleep disruption. It is also worth mentioning that the device can be easily removed during treatment for essential activities such as eating or drinking. All of this makes the system user-friendly and compatible with daily routines, supporting better patient adherence.

Pharmacists, on the other hand, would be instrumental in selecting or preparing the appropriate gel formulations, adapting them to specific treatment needs, and ensuring compatibility with the device. Patients stand to benefit significantly from this approach, as it simplifies treatment regimens. The ability to refill or replace the reservoir extends the usability of the device, potentially reducing overall costs for long-term therapies. The low complexity of the tube design and the use of standard materials such as silicone make this a cost-effective solution compared to more complex commercial drug-delivery systems. The proposed commercial product would combine sterilized tubes and gel formulations packaged together, allowing the user to fill the tube with gel using a syringe. Customized mouthguards, fabricated in dental centers, would complement this system to ensure proper fit, comfort, and safety, especially at night. While there are no directly comparable commercial products currently available that integrate both a reservoir and a mouthguard, the concept aligns with ongoing efforts in personalized medicine and localized drug delivery.

## 5. Conclusions

The findings of this study, including volunteer comfort and the in vivo hydrogel erosion analysis using the tubes, indicated that the proposed silicone tubes (T1, T2) demonstrate significant potential as reservoirs for personalized medical devices. The type of polymer had a significant role in the erosion rate, suggesting that different polymers could be used for specific applications. The tubes protect the formulation from the washing-out effect of saliva, which will prolong the residence time of the drug formulation inside the oral cavity; this will improve the therapeutic efficacy for several treatment applications locally or systemically of this new drug delivery in the future. 

Building on these findings, the development of the reservoir design will incorporate participant feedback from this study to better fit the gum anatomy, whether for the upper teeth or lower gum as an alternative application site. Modifications to the number and size of orifices in the reservoir, along with careful selection of the gel-forming polymer, are key to achieving a more controlled drug release profile. Additionally, integrating this reservoir system with a customized mouthguard device will enhance patient comfort and safety, making it a more practical and user-friendly solution for extended and personalized therapies.

## Figures and Tables

**Figure 1 pharmaceutics-17-01095-f001:**
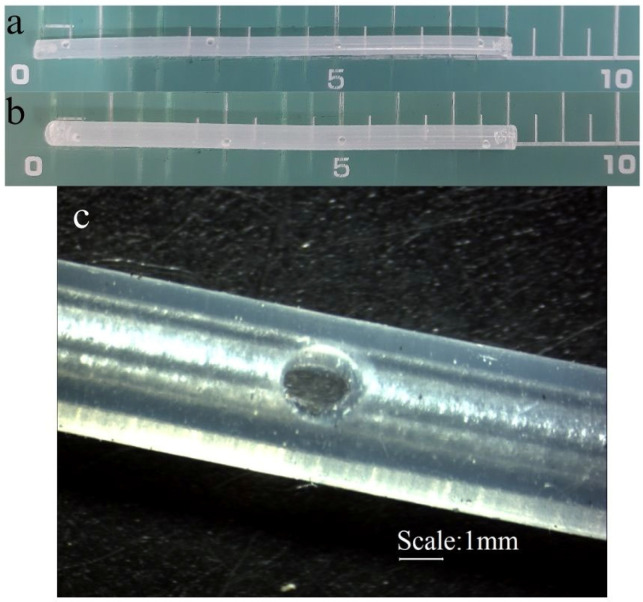
T1 (**a**) and T2 (**b**) silicon tubes used as reservoirs, and (**c**) a microscopic image of an orifice in the T2 tube.

**Figure 2 pharmaceutics-17-01095-f002:**
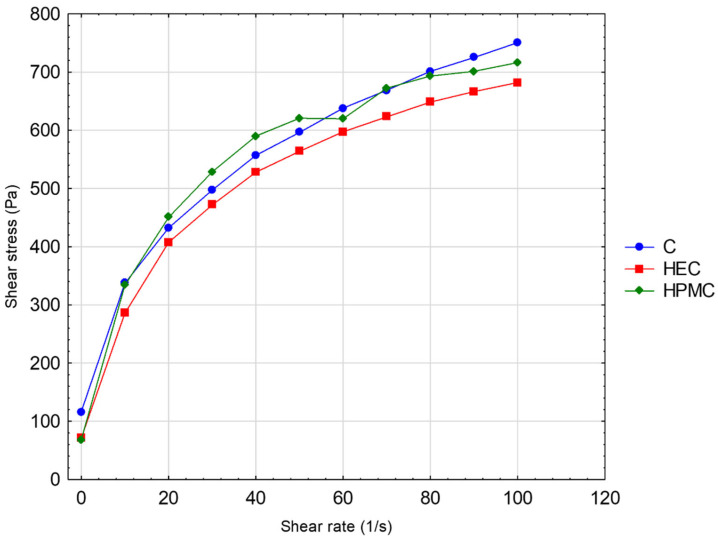
Rheograms of hydrogels: C-carbomer, HEC-hydroxyethylcellulose, HPMC-hypromellose.

**Figure 3 pharmaceutics-17-01095-f003:**
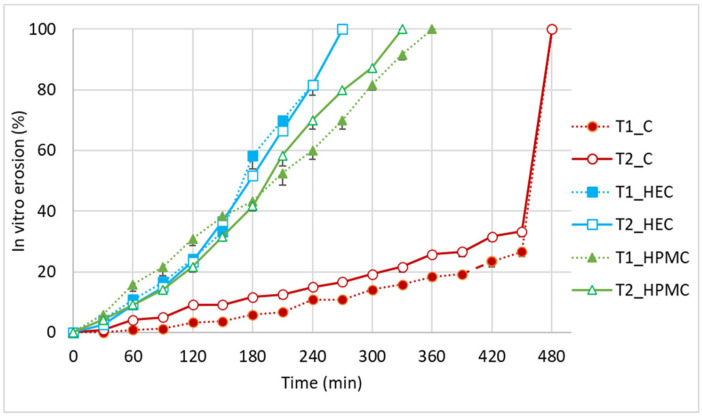
In vitro erosion profiles of the HPMC and HEC gels in tubes T1 and T2.

**Figure 4 pharmaceutics-17-01095-f004:**
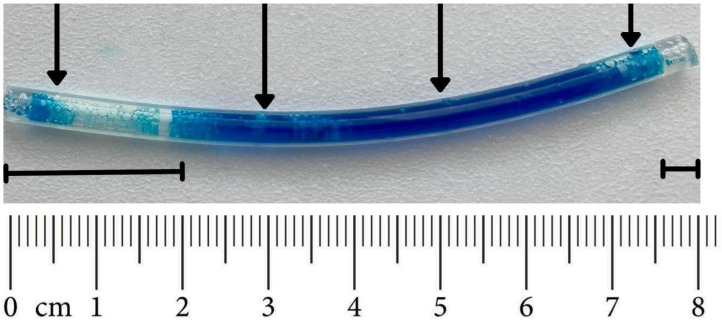
HPMC hydrogel remaining in the tube (T2) after 4 h of application in a volunteer: 30% elution was estimated based on the measurement of empty spaces within the tube (the length of such sections is shown by horizontal lines above the scale). Black arrows indicate the locations of the orifices.

**Figure 5 pharmaceutics-17-01095-f005:**
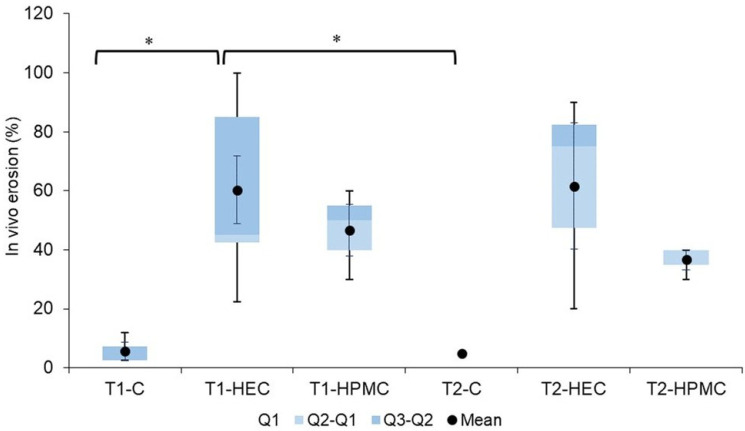
In vivo percentage erosion of hydrogels (HEC, HPMC, C) in tubes T1 and T2 after 4 h of application (*n* = 3; for T1-HEC *n* = 7; * Statistically significant difference; confidence intervals are in Appendix A).

**Figure 6 pharmaceutics-17-01095-f006:**
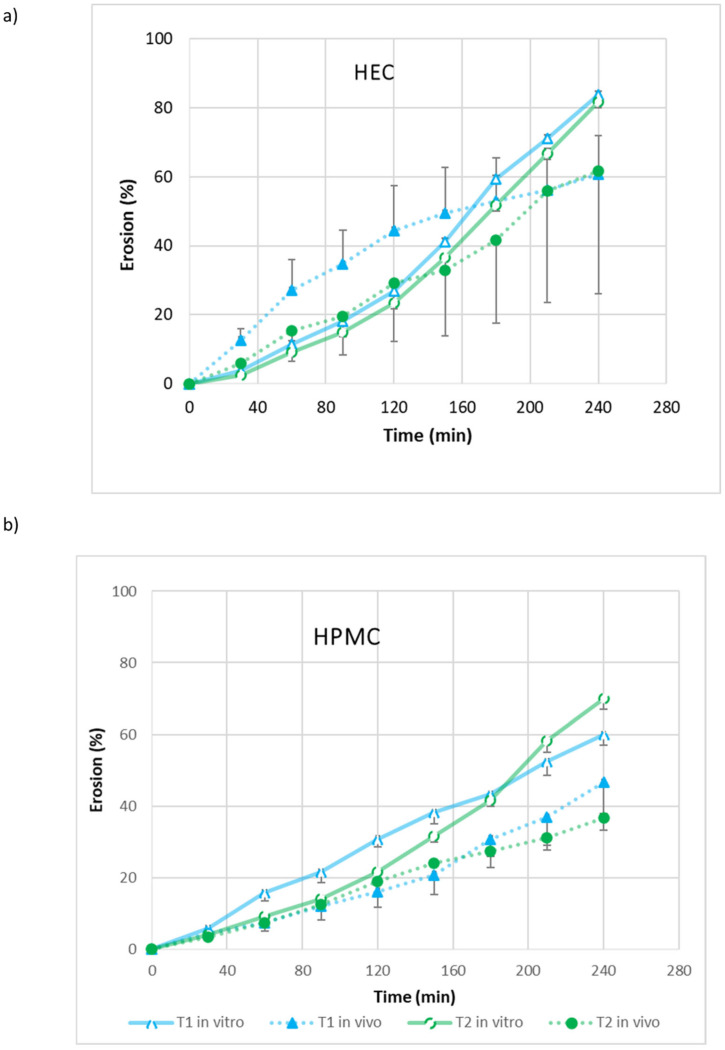
In vitro and in vivo erosion profiles of HEC gel (**a**) and HPMC gel (**b**) over 4 h in tubes T1 and T2 (mean ± sd; *n* = 3, and T1-HEC in vivo *n* = 7).

**Table 1 pharmaceutics-17-01095-t001:** Physicochemical properties of the hydrogels compounded with hypromellose (HPMC), hydroxyethylcellulose (HEC), and carbomer (C).

	HPMC	HEC	C
Polymer concentration [%*w*/*w*]	3.0	4.0	1.5
pH value	5.51	5.98	5.85
Viscosity [Pa·s] at shear rate 20 s^−1^	20.9 ± 1.2	20.8 ± 0.2	21.0 ± 0.5
Hardness (g)	245.6 ± 4.7	248.6 ± 2.7	392.5 ± 7.1
Adhesiveness [g·s]	−602.4 ± 8.4	−600.2 ± 19.2	−619.9 ± 20.5
Cohesiveness [g·s]	482.7 ± 6.3	485.2 ± 10.7	683.4 ± 34.9

## Data Availability

Data supporting the reported results can be obtained from the Authors upon request.

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
