# Peer review of "Prolonged Gel Delivery to Oral Cavity from a Silicone Tube: In Vivo Assessment"

_pharmaceutics, 2025, doi:10.3390/pharmaceutics17091095_

Round 1
Reviewer 1 Report
Comments and Suggestions for Authors
The present manuscript entitled “Prolonged Gel Delivery to Oral Cavity from a Silicone Tube: In Vivo Assessment” reported to develop a silicon tube which can be installed in the oral cavity to provide prolonged release for local or systemic action. My specific comments are as follows;
- It is not clear that how these devices differ from the existing devices like Bocaliner™ or drug-eluting mouth guards. Please clearly articulate the innovation and how this reservoir design advances the field beyond current literature.
- The study involved only six human participants and three additional volunteers for direct gel application. While valuable for a pilot study, this small sample size undermines the robustness of comfort and in vivo erosion data. The authors should include a justification for this sample size and acknowledge its limitations in the discussion.
- The erosion percentage was determined via visual inspection and photography, which introduces subjectivity. Quantitative imaging analysis (e.g., using ImageJ or automated area measurement) could enhance accuracy and reproducibility.
- The study uses carbomer, HPMC, and HEC. The rationale behind the specific concentrations chosen for each (1.5%, 3%, and 4% respectively) needs to be better justified—especially since viscosity was controlled, but mechanical properties varied.
- The study presents strong IVIVC data for HPMC and moderate for HEC. However, this is underexplored in the discussion. The authors should provide more commentary on the relevance of this correlation and its predictive utility in product development.
- While initial comfort feedback is promising, the impact on critical parameters like speech and oral function should be discussed more thoroughly, particularly for long-term use (e.g., overnight applications). How does the prototype compare to other intraoral devices?
- Although the study uses placebo gels, the practical implication lies in drug delivery. A future direction or proof-of-concept inclusion of a model drug (e.g., caffeine, chlorhexidine) would significantly enhance translational relevance.
Minor observations
- The manuscript would benefit from professional language editing. Examples:
- “Despite of the similar viscosity” → “Despite similar viscosity”
- “what is caused by the size of devices” → “which is caused by the size of devices”
- Please define HEC, HPMC, and C in the abstract at first mention.
- The figure and description should better explain how the orifices are distributed and oriented in the oral cavity. Is there a directional flow preference?
- Exact p-values are mentioned but confidence intervals for key comparisons (e.g., erosion rates) should be added to improve statistical interpretation.
- Figures 4 and 5 would benefit from the inclusion of scale bars and clearer labels. Additionally, bar charts in Figure 5 could include error bars and significance stars where applicable.
- "Applicated" (used in line 202) is incorrect; consider replacing with "applied".
- Use consistent formatting for units (e.g., “Pa s” should be written as “Pa·s”).
- Consider uploading raw datasets or photographs as supplementary files, especially since erosion was visually determined.
- A few references are cited as "[6]" or "[13,14]" without context. Ensure that each reference supports the claims made, and follow MDPI formatting guidelines.
The manuscript would benefit from professional language editing. Examples:
-
- “Despite of the similar viscosity” → “Despite similar viscosity”
- “what is caused by the size of devices” → “which is caused by the size of devices”
Reviewer 2 Report
Comments and Suggestions for Authors
Dear Authors,
It is an interesting paper, which provides in vivo results of controlled gel release in the oral cavity using a simple construction: a silicon tube with two different internal diameters and four orifices. Three different polymer-based gels (specifically, HPMC, HPC, and PAA) were used in different concentrations to obtain close rheograms. Despite the similarity in rheology, gels exhibited different properties when tested with texture analysis (such as hardness and cohesiveness) and in vivo gel release profiles. The perception of the medical device and gel release in vivo (based on the loaded ingredients) was assessed via the volunteer questioning.
It is a well-written paper with interesting and valuable information. The described medical device and the way of the gel release are opening the avenue for a new technological platform for oral drug delivery. The idea is impressive due to the device's simplicity and expected robustness.
There are a few suggestions that can be used for the improvement of the manuscript.
- HEC and HPMC are non-ionizable polymers, while the carbomer (PAA) is.
Please consider the relation between the pKa of PAA, pH of the media, and the presence of Mg2+ and/or Ca2+ in the dissolution and in vivo media. - Please mention the taste thresholds for ingredients incorporated into the gel.
- I guess it would be better to describe the TA method separately.
Kind regards
Round 2
Reviewer 1 Report
Comments and Suggestions for Authors
The authors have addressed all the concerns raised during the original submission.